# Structural Characterization, Cytotoxicity, and the Antifungal Mechanism of a Novel Peptide Extracted from Garlic (*Allium sativa* L.)

**DOI:** 10.3390/molecules28073098

**Published:** 2023-03-30

**Authors:** Shuqin Li, Yajie Wang, Jingna Zhou, Jia Wang, Min Zhang, Haixia Chen

**Affiliations:** 1Tianjin Key Laboratory for Modern Drug Delivery & High-Efficiency, School of Pharmaceutical Science and Technology, Tianjin University, Tianjin 300072, China; 2College of Food Science and Bioengineering, Tianjin Agricultural University, Tianjin 300384, China; 3State Key Laboratory of Nutrition and Safety, Tianjin University of Science & Technology, Tianjin 300457, China

**Keywords:** garlic antifungal peptide, structural identification, drug resistance, membrane destruction, transcriptomic analysis

## Abstract

Garlic (*Allium sativa* L.) is a traditional plant with antimicrobial activity. This study aimed to discover new antifungal peptides from garlic, identify their structure, and explore the antimicrobial mechanism. Peptides were separated by chromatography and identified by MALDI-TOF analysis. Structure and conformation were characterized by CD spectrum and NMR analysis. Mechanism studies were conducted by SEM, membrane depolarization, and transcriptomic analysis. The cytotoxicity to mammalian cells as well as drug resistance development ability were also evaluated. A novel antifungal peptide named NpRS with nine amino acids (RSLNLLMFR) was obtained. It was a kind of cationic peptide with a α-helix as the dominant conformation. NOESY correlation revealed a cyclization in the molecule. The peptide significantly inhibited the growth of *Candida albicans*. The mechanism study indicated that membrane destruction and the interference of ribosome-related pathways might be the main mechanisms of antifungal effects. In addition, the resistance gene *CDR1* for azole was down-regulated and the drug resistance was hardly developed in 21 days by the serial passage study. The present study identified a novel antifungal garlic peptide with low toxicity and provided new mechanism information for the peptide at the gene expression level to counter drug resistance.

## 1. Introduction

Conventional antimicrobials are becoming increasingly ineffective against some of the infectious diseases, which is becoming a public health problem. Now, the post-antibiotic era urges the development of novel therapeutics, as well as the need to cope with the drug resistance problem that current antimicrobials arouse [1]. Fungal infections continue to be a significant problem for infectious disease control worldwide, particularly for patients with compromised immune function [2]. *Candida* species, especially *Candida albicans*, are the leading cause of invasive mycotic disease among fungal infections. However, despite the detrimental impact of fungi on global health, the development of novel antifungal components is relatively slow, with only a few classes of drugs available for use, such as polyenes, azoles, and echinocandins, and the efficacy of these agents is compromised by the development of drug resistance in pathogen populations [1,2,3,4]. The global health problem has spurred a worldwide mandate to identify potentially effective alternatives.

Antimicrobial peptides (AMPs) are attractive for drug development attempts due to their broad-spectrum antibacterial and antifungal characteristics. As essential components of the innate immune system, AMPs generally have a high selectivity between host and pathogen cells, and the development of resistance against them is rarely reported [5]. The fundamental characteristic of AMPs is the cationic feature that enables them to interact directly with cell membranes, causing an increase in membrane permeability, endowing them with direct interaction with cell membranes, and provoking an increase in membrane permeability [6]. Antifungal peptides (AFPs) have captured the attention of researchers for their benefits for human health and crops against fungal infections with enhanced drug resistance. Apart from the different modes of action of AFPs, it also meets the desired characteristics for drug development with low toxicity toward mammals and is unlikely to develop resistance [7]. AFPs have been identified in many organisms. Plant food as a source of AFPs allows the development of food-grade candidates suitable for general application, even in vegetarian and vegan products. The advantage of plants also lies in the fact that plants do not have an immune system when they are exposed to a large number of pathogenic fungi [8]. Until now, over 2800 AMPs from natural sources are currently registered on the Antimicrobial Peptide Database (APD) [9], among which a few of them are AFPs. However, the drawbacks of AMPs, such as low stability and activity in the body environment and low specificity toward pathogens, hinder the development of AFPs [10]. Therefore, the discovery of new peptides with more ideal characteristics is an urgent request for antibiotic development.

Garlic (*Allium sativa* L.) is a bulbous flowering plant that belongs to the onion genus *Allium* within the lily family *Liliaceae*, and it is a predominant horticultural crop originating from central Asia [11]. Garlic has long been widely used in traditional medicine owing to its numerous activities, such as anti-tumorigenesis, anticarcinogenic, anti-atherosclerotic, antithrombotic, antidiabetic, antiplatelet aggregation, and anti-inflammatory activities [12,13]. The antifungal activity of garlic products was first established in 1936. Since then, several studies have verified the fungicide activity of different components, especially sulfur-containing compounds [12,13,14,15], and ajoene [(E, Z)-4,5,9 Trithiadodeca 1,6,11 Triene 9-oxide], which was stable in water and could be obtained by chemical synthesis, was demonstrated with diverse bioactivities including antifungal properties [16]. Garlic extracts and garlic oil, for instance, were also extensively verified with antifungal activity against *Candida* spp. Garlic oil was constituted by a number of linear sulfur-containing volatile compounds where diallyl disulfide (DDS) and diallyl trisulfide (DTS) were the most abundant volatile compounds [14,17,18]. However, AFPs from garlic were rarely reported. In our previous study, potent AMPs were separated from traditional Laba garlic [19], which indicated that garlic might be a promising source for AFP isolation.

Therefore, the present study aimed to isolate antifungal peptides from garlic, characterize the structure and conformation, study the cytotoxicity, and evaluate the antimicrobial activities against the fungal pathogen *Candida albicans*. In addition, the anti-drug resistance properties of the peptide were also evaluated. To achieve these goals, scanning electron microscopy, the membrane depolarization assay, and transcriptomic analysis were utilized to explore the underlying function mechanism of the antifungal peptides, and several genes related to drug resistance were also analyzed for their implications in the overall activity evaluation.

## 2. Results and Discussion

### 2.1. Peptide Isolation and Identification

Considering the anionic characteristic of the negatively charged cell membrane, the peptide with the cationic feature was intended to be separated. DEAE Sepharose fast flow chromatography used an anionic exchange column; thus, the cationic fraction was supposed to be separated in the unbounded fraction. By DEAE Sepharose fast flow chromatography separation, an elution peak was observed in unbounded fraction when eluted with PBS (Figure 1A). The obtained fraction was further identified by MALDI-TOF MS/MS de novo sequencing, two cationic peptides with a large portion of hydrophobic residues (>50%) were sequenced and identified, and they were supposed to have antimicrobial activity according to the common characteristics of AMPs [20]. The sequence details of the two peptides were listed as *N*-terminal-Leu- Met-Leu-Leu-Met-Leu-Phe-Arg-*C*-terminal (LMLLMLFR) and *N*-terminal-Arg-Ser- Leu-Asn-Leu-Leu-Met-Phe-Arg-*C*-terminal (RSLNLLMFR). Then, the peptides were synthesized by GenScript (Nanjing, China) with a purity ≥95% and were subjected to antimicrobial ability evaluation. Results of antifungal and antibacterial activity evaluation indicated that both peptides had no inhibition activity toward *Escherichia coli* and *Staphylococcus aureus*, but peptide RSLNLLMFR (NpRS) showed antifungal activity in coping with *C. albicans* and thus was subjected to further kinetic and mechanism analysis. The sequence of the synthesized peptide was further verified by MALDI-TOF MS/MS analysis (Figure 1B). NpRS had an amphipathic structure, and its molecular weight was 1149.41 Da. The isoelectric point of the peptide was at pH 12.1 and was cationic in the physiological environment, which might be owed to the basic amino acid arginine. According to previous studies, the cationic properties were closely correlated with the antimicrobial activity of the peptides, which was due to the negatively charged cell membrane [21]. It was also reported that amino acid Arg residues with cationic charges facilitated a vital means of attracting the peptides to the target membranes, and the hydrogen bonding properties provided its interaction with negatively charged surfaces, such as phosphatidyl glycerol phospholipid head groups [22]. In sum, the constitution of amino acids strongly affected the potent antimicrobial activity and specificity to the microbial target. In addition, the amphipathic nature of the peptides also showed a high affinity for microbial membranes. Therefore, the particular amphipathic structure and Arg residues in NpRS might play vital roles in its antifungal activity.

### 2.2. Antifungal Activity of NpRS

The antifungal activity of NpRS against *C. albicans* was qualitatively and quantitatively assessed by the values of minimum inhibition concentration (MIC) and the time-kill kinetics assay. The MIC of NpRS against *C. albicans* was 0.27 mM. The time-kill kinetics assay was performed with fungi strains to determine the mode of action of NpRS on the growth of *C. albicans*. At certain time intervals, aliquots were taken and determined by OD_600_. According to the results (Figure 1C), NpRS exhibited significant fungicide activity (*p* < 0.05) and extended the lag phase of *C. albicans* at the MIC concentration. Though the lag phase of the positive control, fluconazole, was longer, its concentration was over 6 times that of NpRS. Moreover, NpRS inhibited the fungi growth in 36 h at 2×  MIC without a detected fungi intensity increase, which indicated that NpRS was an effective antifungal agent.

### 2.3. Structure and Conformation Characterization of NpRS

The structure detail is strongly associated with the antimicrobial activity of peptides, including the molecular length, amino acid composition, hydrophobicity, net charge, and secondary structure [21]. According to PepCalc, the net charge of NpRS at pH 7 was +2, which facilitated the preliminary electrostatic interaction with the anionic membrane. The results from the CD spectrum confirmed that the α-helical structure was the dominant configuration, and the proportion of the α-helix was more than 50% both in pH 5 and pH 7 aqueous solutions, which indicated that the structure of the peptide was comparatively stable when it was applied in the acidic environment of the human body (Figure 2A,B). Naturally occurring antimicrobial peptides can be divided into several categories, and one of the predominant classes of these peptides is linear with an α-helical structure [10]. A previous study reported that an antifungal peptide derived from hemocyanin could permeabilize through fungal membranes due to the α-helical structure and result in cell death [23]. Therefore, the amphiphilic characteristic with a cationic charge that combined with the α-helical structure might be the indispensable factor for NpRS’s antifungal activity. To further study the spatial conformations of NpRS, a 2D NOESY analysis was conducted (Appendix A). According to the result (Figure 2C), the inner structure of the hydrogen molecules correlated spatially in the positions of C38 and C27. Such a correlation might be the basis for α-helical structure formation. Such a conformation might also be beneficial for the organization of hydrophobic and cationic amino acids in discrete sectors, and thus laid the foundation for the amphiphilic structure of the molecule and facilitated the interaction with the hydrophobic target membrane. In addition, cyclization has been recognized as an effective approach to improve the efficacy of the peptide and small-molecule therapeutics [24]. The formation of cyclization may be beneficial for metabolic stability and proteolytic resistance [25]. Therefore, the structural characteristics of NpRS might be strongly correlated with its high antifungal effect.

### 2.4. Antifungal Mechanism of NpRS

#### 2.4.1. SEM and Membrane Potential Studies for *C. albicans* after NpRS Treatment

Scanning electron microscopy (SEM) is a useful technique to visualize the effect of peptides on pathogens. After the drugs’ treatment, cell membrane wrinkles, distortions, and cell pores in the cell membrane morphology usually indicate damage to the cell membrane and cell death [26]. To investigate the possible mechanism of the peptide, the morphology of *C. albicans* was observed by using SEM. According to the results (Figure 3A,B), NpRS at 1× MIC concentration affected *C. albicans*’s cell surface morphology when compared with the control group. Three hours after the treatment, the cell surface was depicted with roughness and shrinkage, and several pores were formed on the surface. In addition, cell debris was also observed under the microscope, which indicated that NpRS might destroy the cell wall and membrane, thus affecting the cell permeability and promoting cell death.

The alteration in membrane potential was also determined to figure out the mode of action toward *C. albicans*. DiBC4(3) is a voltage-sensitive fluorescent probe and is usually used to observe the alterations in membrane potential. The dye easily enters the depolarized cells and the fluorescence enhances [27]. Figure 3C shows the changes in fluorescence intensity of *C. albicans* after NpRS treatment at different concentrations (0× MIC, 0.5× MIC, 1× MIC, 2× MIC). Compared with the control group, the fluorescent intensity was significantly magnified in a dose-dependent manner, which reflected the induction of membrane depolarization of *C. albicans*. Cell membrane depolarization was regarded as an important sign of cell membrane damage [28]. Therefore, combined with the results of SEM together, membrane damage might be one of the action mechanisms of NpRS against *C. albicans*.

#### 2.4.2. Transcriptomic Analysis

To gain insight into potential molecular pathways for the NpRS antifungal mechanism, transcriptomics analysis of *C. albicans* was performed between the normal control and the NpRS-peptide-treated strains. After the treatment, significant gene expression differences were observed (Figure 4A). Compared with the normal control group, 1850 genes were identified as differentially expressed genes (DEGs), in which 1043 genes were up-regulated and 807 genes were down-regulated (|log2 (FC)| ≥ 2). Based on the possible membrane destruction fungicide mechanism of NpRS, ergosterol-related gene expressions were analyzed. Membrane composition sterols (ergosterol in particular) affect membrane functions, such as efflux pumps, and *ERG11* and *ERG3* are the genes encoding rate-limiting enzymes in ergosterol synthesis and involved in sterol-related synthesis pathways [29]. According to the volcano plot of DEGs related to ergosterol (Figure 4B,C), *ERG 3* was significantly decreased by over 4 folds of changes, indicating that membrane synthesis was affected by NpRS treatment. The result is consistent with that of amphotericin B, which significantly down-regulated the gene expression of *ERG3* [30]. In addition, the expression of gene *FAS1*, which is involved in fatty acid biosynthesis and fatty acid metabolism, was also decreased (Figure 4C). The down-regulation of these genes indicated that the cell membrane stability was strongly affected by the treatment of NpRS, and suggested that membrane destruction was one of the antifungal mechanisms for NpRS function [31].

We further analyzed the biological process (BP), cellular component (CC), and molecular function (MF) by GO enrichment analysis. Among BP terms, cellular process, metabolic process, and single-organism process contained the greatest number of DEGs (Figure 4D). In the CC category, the top three GO terms were cell, cell part, and organelle. Interestingly, the membrane part was also one of the leading variant DEGs in CC terms. In MF terms, catalytic activity and binding were significantly affected by the treatment of NpRS, in which the catalytic activity strongly related to protein-accurate synthesis and the binding was strongly associated with important virulence determinants in various infection models [32]. Like the treatments of other known cationic antimicrobial peptides, disruption of the bacterial cell membrane was one of the mechanisms of action of NpRS [33].

KEGG enrichment analysis was conducted to identify the affected biological pathways under the treatment of NpRS. Based on the results (Figure 5), 20 KEGG pathways were enriched. The ribosome was the most significantly down-regulated pathway in the analysis and was followed by carbon metabolism. Previous studies had revealed that ribosome acted as the hub of protein quality control, and its main responsibilities were building amino acid sequences by genetic code and assembling protein polymers using amino acid monomers as the infrastructure [34]. The function of the ribosome was pivotal for microbial protein synthesis and thus essential for fungi’s normal life activity. There were many clinically useful antibiotics that showed bactericidal effects by inhibiting protein synthesis on microbial ribosomes, and the ribosome-targeting agents were thought to be the intriguing direction for new antibiotic development [35]. Combined with the MF analysis in GO enrichment, the catalytic process variant might be owed to the ribosome dysfunction and then resulted in enzyme misfolding or incorrect assembling. Therefore, in the present study, the ribosome-related pathway down-regulation might be another important path for NpRS’s antifungal mechanisms.

#### 2.4.3. Drug Resistance Analysis

Drug resistance was a significant problem for antibiotic treatment. For further detection of the susceptibility toward drug resistance, a serial passage study was conducted. The drug-resistant strain was not obtained by serial passage of *C. albicans* in the presence of sub-MIC levels of NpRS over 21 consecutive days, and the MIC fold of change was no larger than 1 fold of change during the process (Figure 6A). These results usually pointed to a non-specific mode of action, but with accompanying toxicity [36]. The positive control fluconazole soon developed resistance in 8 days where the MIC was 16 times of the initial concentration, and the experiment was suspended due to its poor solubility (Figure 6B). Fluconazole is the most widely used drug for *Candidiasis* infections and mainly targets ergosterol biosynthesis for *Candida* membrane integrity. In fluconazole-resistant strains, the expression level of the *ERG11* gene was increased and was responsible for antifungal resistance [37]. Instead of targeting one single target to interrupt the metabolic reaction and cell growth, the antimicrobial mechanism of AMPs was usually demonstrated as a disruption of the cell membrane. Therefore, the different mode of action was equipped with difficulties in developing drug resistance [21]. The results in our present study further indicated that the mode of action of NpRS might be associated with membrane interaction. Facing azole treatment, several genes related to azole resistance were reported to be increased, including *CDR1* and *CDR2* [30]. *CDR1* and *CDR2* are ATP-binding cassette transporter genes encoding multidrug efflux pumps [38]. In the current study, the transcript of *CDR1* was significantly decreased (*p* < 0.05), indicating that the treatment of NpRS would not induce drug resistance like azole does, and might reduce the susceptibility of drug resistance of azoles (Figure 6C). However, *CDR2* gene expression was also decreased but within two folds of changes. We also explored the synergistic effect between fluconazole and NpRS. The results revealed that FICI was in the range of 0.5–4, which indicated that no synergistic interaction was observed. The reason might be due to the fact that the single target of fluconazole was covered by a larger scale of membrane interaction, which also further sustained the membrane interaction of NpRS. The proposed antifungal mechanism of NpRS against *C. albicans* is summarized in Figure 6E, and the regulation on the gene expression level of NpRS will need further study.

### 2.5. Cytotoxicity of NpRS against Mammalian Cell

The mechanism for the MTT assay is that metabolically active cells would transit the yellow tetrazolium salt MTT to purple formazan crystals. Lower cytotoxicity indicates the membrane selectivity of peptides toward different membrane types, and the selectivity is also important for the clinical application of drugs. Figure 6D shows the survival rates of rat L6 cells under the treatment of NpRS at different concentrations. The results showed that cell viabilities were decreased along with the increase in peptide concentration, indicating dose-dependent toxicity. It was reported that increased antimicrobial activity is usually accompanied with increased mammalian cell toxicity [39]. However, the cell viability was close to 80% even at the highest dose tested, reflecting the viability at MIC levels (between 250 and 500 μg/mL). In a previous cytotoxicity study, a cell viability reduction not more than 30% of the control was considered to have a moderate cytotoxic effect [40]. Therefore, NpRS might have a specific selectivity between different membrane origins and cell toxicity would not be the major concern of NpRS.

## 3. Materials and Methods

### 3.1. Material and Reagents

Garlic bulbs originated from Henan province (China) were purchased from the local market in Tianjin (China). The fungal strain used in this study was *C. albicans* 3147 (ATCC No. 10231) and was obtained from the China Center of Industrial Culture Collection (CICC, Beijing, China). Pepstatin A was purchased from Yuanye (Shanghai, China), Sabouraud dextrose broth was purchased from Solarbio (Shanghai, China), MTT (3-(4, 5-dimethylthiazole-2-yl)-2, 5-diphenyltetrazolium bromide was purchased from Solarbio (Beijing, China), and bis-(1,3-dibarbituric acid)-trimethine oxanol (DiBAC4(3)) was purchased from Bioscience (Shanghai, China). All the other reagents and chemicals were purchased from Concord Technology Co., Ltd. (Tianjin, China) and Jiangtian Technology Co., Ltd. (Tianjin, China) and were of analytical or chromatographical grade.

### 3.2. Extraction and Isolation of NpRS

The isolation of NpRS was conducted according to the previous study with minor modifications [6]. Fresh garlic bulbs were air-dried and ground into a fine powder and defatted with petroleum ether (1:10, *w*/*v*) at room temperature for 1 h. The defatted powder was extracted with a mixture of acidic solution, specifically, 1% (*v*/*v*) trifluoroacetic acid (TFA), 1 M HCl, 5% (*v*/*v*) formic acid, and 1% (*w*/*v*) NaCl in the presence of pepstatin A (1 μg/mL) at a seed/solvent ratio of 1:4 *w*/*v*. The extraction solution was concentrated and desalted on the G-25 column, and further separated by a DEAE Sepharose fast flow column (2.5 × 50 cm), equilibrated with phosphate-buffered solution (PBS) (pH 7.4). To isolate cationic peptides specifically, the unbounded solutions eluted by PBS were collected for further analysis.

### 3.3. Peptides’ Identification by MALDI-TOF MS/MS Analysis

Matrix-assisted laser desorption ionization (MALDI)-time of flight (TOF)/TOF MS has been considered one of the state-of-the-art characterization techniques for the investigation of natural molecules such as proteins, peptides, oligosaccharides, and synthetic small molecules of low molecular weight. The peptide in the fractioned solution was characterized using an UltrafleXtreme MALDI-TOF/TOF MS (Bruker Daltonics, Bremen, Germany) operated at 7.5 kV and a time delay of 90 ns in positive (reflectron) ion mode. The fraction was added to a matrix solution (1:1, *v*/*v*) of α-cyano-4-hydroxycinnamic acid. Then, the mixture was placed on the steel plate MTP Anchor Chip TM TF 600/384 (Bruker Daltonics, Bremen, Germany) by a pipette. The turbo-molecular pumps were employed to maintain the source of ion discharge. The analysis method for MS1 was the RP PepMix (700–3500 Da) and the analysis for MS2 was lift-positive. Peptides were identified utilizing the standard Peptide Calibration Standard II (Bruker Daltonics, Bremen, Germany).

The peptide mass fingerprint obtained by MS2 analysis on MALDI-TOF/TOF was edited by Flex Analysis 3.4 software (Bruker Daltonics, Bremen, Germany) and saved as peak list files in the mgf. extension. For the next step of the analysis, peptide sequence identification was performed in the MASCOT database with the following parameters: taxonomic group “*Lilium*”, no cleavage by enzyme, peptides mass maximum variation of 0.5 Da, allowing no cleavage loss, peptide +1 charge, and instrumentation MALDI-TOF/TOF. The partial amino acid sequence of the peptide was analyzed for its sequence homology using an NCBI-BLAST analysis [41,42]. The identified peptides were synthesized by GenScript Corporation (Nanjing, China) with >95% purity, and the sequences were verified by MALDI-TOF MS/MS. The peptides were subjected to further analysis.

### 3.4. Antifungal Activity Evaluation by Agar Diffusion Assay

The antifungal activities of peptides were screened by agar well diffusion methods, which followed our previous study with minor modifications [19,43]. Specifically, 100 μL of *C. albicans* (1 × 10^7^ cfu/mL) at the exponential-phase was well spread on SD agar plates. Holes of agar were created by sterile pipette tips. An amount of 40 μL of peptide solution with a concentration of 5 mg/mL was added in each well. Fluconazole and sterile water were used as the control. Plates were incubated at 37 °C for 24 h. The peptide with the inhibition zone was identified with antifungal activity.

### 3.5. Minimum Inhibition Concentration (MIC) Assay

The MIC was determined to identify the peptide with antifungal activity [19,37]. Fluconazole and sterile water were used as positive and negative controls, respectively. The peptides (initial concentration 17 mM) and fluconazole (initial concentration 26 mM) were dissolved in SD broth with different concentrations by the micro-dilution method. The solutions were added into the 96-well plate with final fungal suspensions at 2.5 × 10^5^ c.f.u/mL. The plate was incubated at 37 °C for 24 h. MIC was defined as the minimum concentration with no fungal growth when observed with naked eyes. The experiment was carried out in triplicates.

### 3.6. Time-Kill Kinetics Analysis

Time-kill kinetics analysis was conducted according to our previous study with minor modifications [19]. Briefly, *C. albicans* cells in the exponential phase (2.5 × 10^5^ c.f.u/mL) were incubated with different dosages of NpRS (1× MIC, 2× MIC) and cultivated for 36 h at 37 °C. Two-hundred-microliter aliquots were detected by a microplate reader at OD_600_ at certain time intervals. Fluconazole was used as the positive control, and the blank control was treated with an equal amount of cultivation medium. Each group was conducted in triplicates.

### 3.7. Synergetic Interaction Analysis with Fluconazole

Chequerboard microdilution studies were conducted as previously reported [44]. The concentrations of NpRS ranged from 1/8 to 4× MIC. The inoculum was performed with the method for the broth microdilution assay. Results were observed visually as no fungal growth in the well. Fractional inhibition concentration index (FICI) was used to quantify the synergetic interactions. Specifically, FICI was calculated as FICI = (MIC_in combination_/MIC_alone_) _Fluconazole_ + (MIC_in combination_/MIC_alone_) _NpRS_. The FICI data were interpreted in the following way: FICI ≤ 0.5 = synergy, FICI > 0.5–4 = no interaction, and FICI > 4.0 = antagonism.

### 3.8. Secondary Structure Analysis by Circular Dichroism (CD)

The secondary structures of the NpRS at different pH environments were detected according to our previous study with minor modifications and detected by ALX-300 (BioLogic, China) [11]. Briefly, NpRS (0.5 mg/mL) was dissolved in PBS and coordinated by HCl (0.5 M) solution to different pHs (pH 5, pH 7). The solution was added to a quartz cell. The scanning range of wavelengths was set at 190–260 nm, the bandwidth was 1.0 nm, the path length was 100 nm, and the scan rate was 100 nm/min. The spectra were averaged over three scans and PBS solution with different pHs was used as the blank control.

### 3.9. Spatial Structure Analysis by Nuclear Magnetic Resonance (NMR)

NpRS (10 mg) was completely dissolved in deuterium oxide (D_2_O) (99.8% D, Cambridge Isotope Laboratories Inc., Andover, MA, USA) and inspected with an Avance III 400 MHz NMR spectrometer (Bruker Corporation, Switzerland) to obtain its 1D and 2D NMR spectra. The chemical shifts were expressed in ppm, and the signal of deuterated water was fixed at 4.7 ppm.

### 3.10. Scanning Electron Microscopy (SEM) Analysis

The morphological alterations after exposure to NpRS were observed by field emission scanning electron microscopy (Apreo S LoVac, Czech) [19]. *C. albicans* cells were incubated in SD broth at 37 °C and the cell concentrations were set at 2.5 × 10^5^ c.f.u/mL. The prepared cell solutions were treated with different concentrations of peptide (0× MIC, 1× MIC). All samples were incubated at 37 °C for 3 h. After the treatment, the cells were collected by centrifugation at 1500× *g* for 10 min and washed with 0.1 M PBS. The cells were fixed with 2.5% glutaraldehyde and then dehydrated with gradient ethanol (10%, 30%, 50%, 70%, 90%, 100%). All the samples were subjected to gold coating and observed under an electron microscope.

### 3.11. Membrane Depolarization Analysis

The method was performed before following a previous study with minor modifications [45,46]. *C. albicans* was grown in SD broth at 37 °C to a density 1 × 10^7^ c.f.u/mL. Cells were exposed to different concentrations of NpRS (0× MIC, ½× MIC, 1× MIC, 2× MIC) for 5 min, and 1 mM DiBAC4(3) was added to reach a final concentration of 1 μM and mixed thoroughly and incubated for another 30 min in the dark. The fluorescence was tested by a spectrofluorometer (RF-6000, Shimadzu, Japan) at the wavelengths of 490 nm and 516 nm for excitation and emission, respectively. The excitation and emission slit widths were 3 and 5 nm.

### 3.12. Transcriptome Analysis

*C. albicans* cells were cultivated for 24 h with continuous shaking to the exponential phase and were diluted (2.5 × 10^5^ c.f.u/mL). Subsequently, cells were exposed to a subinhibitory concentration (0.135 mM) for 24 h in triplicates. After exposure, the cells were immediately pelleted by centrifugation at 3000× *g* for 10 min in 1 mL aliquots. The precipitants were collected and immediately frozen in liquid nitrogen at −80 °C until total RNA isolation. Cells without NpRS treatment were used as the blank control group and cultivated in the same condition. Total RNA was isolated. After quantity and quality monitoring, 18 cDNA libraries were sequenced using the Illumina Novaseq 6000 platform in the Biomarker company (Beijing, China). For identification of differential expression genes (DEGs), genes with an adjusted |log2 (Fold change) | ≥ 2 and FDR < 0.01 were assigned as significantly differentially expressed. Gene Ontology (GO) enrichment analysis was performed. GO annotation is based on the significant enrichment of GO functions to analyze DEGs and related gene modules for bioinformatics analysis. GO covers three aspects of biology: cellular components, molecular functions, and biological processes. The KEGG database (https://www.kegg.jp/ (accessed on 21 September 2022)) can systematically classify and annotate the metabolic pathways of genes and can be used to analyze gene expression information at a general level.

### 3.13. Drug Resistance Evaluation by Sequential Passaging

Drug resistance was determined according to a previous study with minor modifications [36]. *C. albicans* cells were cultivated for 24 h with continuous shaking to the exponential phase and diluted 1:5000 in fresh SD broth. An amount of 50 μL of aliquots of diluted cells was added in a 96-well plate in the presence of different concentrations of NpRS or fluconazole by broth microdilutions. Cells were incubated at 37 °C for 24 h and the MIC was recorded. Cells at a subinhibitory concentration were diluted 1:5000 in fresh medium and were passaged to another round of tests in the presence of NpRS or fluconazole. The process was performed daily for 21 days. MIC fold changes were recorded.

### 3.14. Cytotoxicity Assay against Mammalian Cell

The MTT (3-(4,5-dimethylthiazole-2-yl)-2,5-diphenyltetrazolium bromide) assay was performed to evaluate the cytotoxicity effect of NpRS [47]. Briefly, L6 cells were plated in a 96-well plate at a concentration of 1 × 10^5^ cells per well supplemented DMEM medium (10% FBS and 100 units/mL of penicillin/streptomycin). Cells were treated with different dosages of NpRS (31.25–500 μg/mL) and cultured in 5% CO_2_ at 37 °C. After overnight incubation, 50 μL of MTT solution (2 mg/mL in PBS) was added to each well and further incubated for 4 h. The supernatant was removed from each well and 100 μL of 100% DMSO was added to dissolve the blue formazan product. The absorbance was read at 490 nm, and the cell viability was expressed as AT/AC × 100%, where AT and AC are the absorbances of treated and control cells, respectively.

### 3.15. Statistics

All the data analysis was performed by GraphPad Software. The results were expressed as the means ± standard deviation in all experiments. The hydrophobicity and net charge were calculated by a Peptide Property Calculator (http://pepcalc.com/ (accessed on 20 March 2022)). All the experiments were conducted in at least triplicate.

## 4. Conclusions

Based on the established antimicrobial activity of garlic, this study discovered a novel antifungal peptide NpRS from garlic with nine amino acids (RSLNLLMFR). The study also elucidated the structure characteristics as well as the antifungal mechanisms of the peptide from a new perspective. Specifically, the study revealed that the amphiphilic characteristics facilitated the primary electrostatic interaction with the anionic cell membrane, and the α-helix structure was the dominant conformation. The inner-molecule cyclization revealed by the NOESY correlation indicated that the correlation might be the basis for α-helix formation and also might be the main core responsible for antifungal activity. Membrane destruction was the main function mechanism for NpRS. In addition, the transcriptomic analysis revealed that the interference of protein synthesis by the ribosome was another way for NpRS’s antifungal function. NpRS was hardly developed with drug resistance in a long time range, which indicated that the target of the peptide was not a specific protein receptor but aimed at the whole membrane. Furthermore, NpRS might attenuate the resistance caused by azole drugs by the downregulation of the resistance gene *CDR1*. However, an in vivo study would be necessary for a systematic evaluation of the antifungal activity, and this is also a necessary step for most of the antimicrobial agents for future commercial development. In addition, the present study only presents one antifungal peptide from garlic, so more possible peptides with diverse activities are expected to be separated from garlic. Therefore, the present study identified a novel antifungal peptide from garlic with a novel antifungal mechanism. The research would be beneficial for future drug development, and also important for garlic market value improvement.

## Figures and Tables

**Figure 1 molecules-28-03098-f001:**
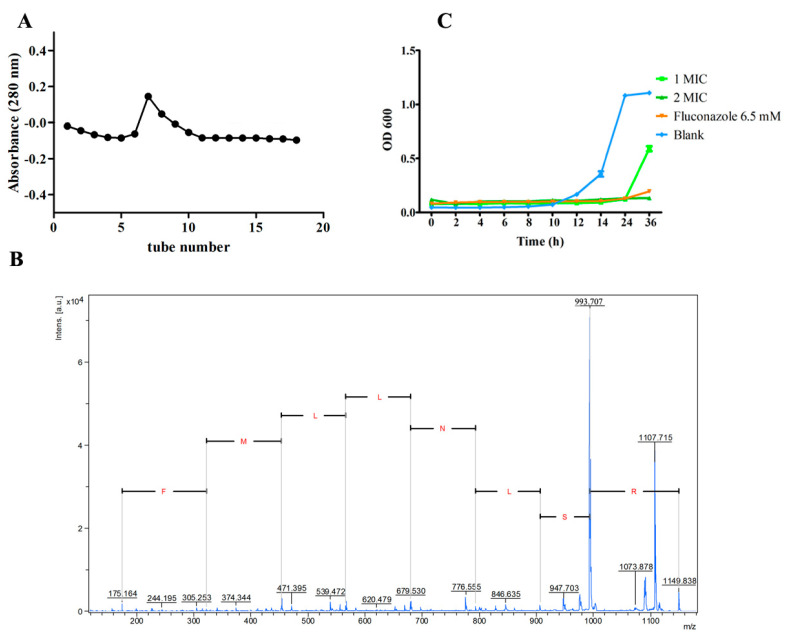
Isolation, identification, and antifungal activity of NpRS. (**A**) DEAE−fast flow chromatography elution curve of peptide fraction eluted with PBS for the unbounded fraction; (**B**) MALDI−TOF MS/MS spectra of NpRS; (**C**) Growth inhibition curve by different concentrations of NpRS. All data were expressed as the mean values of triplicate ± standard deviation.

**Figure 2 molecules-28-03098-f002:**
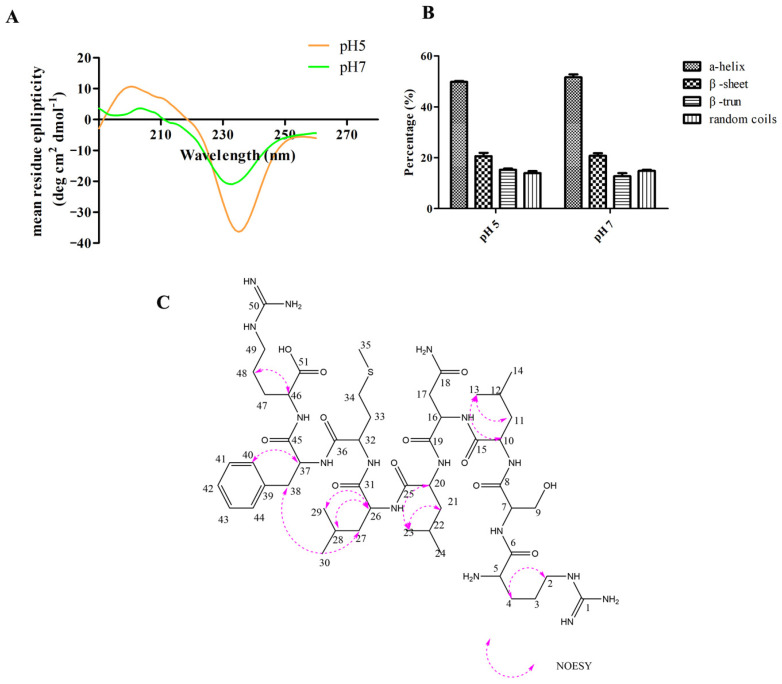
Structure analysis of NpRS. (**A**) CD spectrum of NpRS in different pH environments; (**B**) The ratio of the secondary structure of NpRS in different pH environments; (**C**) NOESY correlation of NpRS. All data were expressed as the mean values of triplicate ± standard deviation.

**Figure 3 molecules-28-03098-f003:**
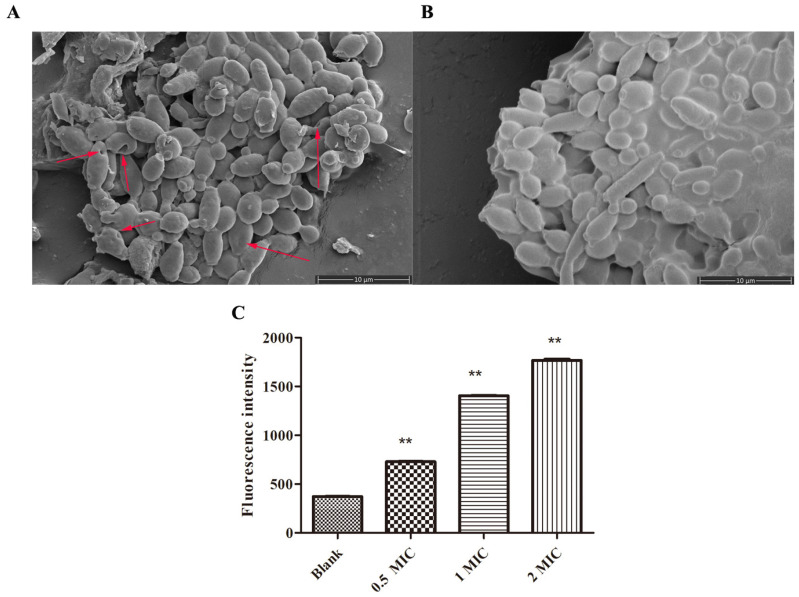
Antifungal mechanism of NpRS. (**A**) Scanning electron microscope image of *C. albicans* under the treatment of NpRS (1×MIC) after 3 h of incubation; (**B**) Scanning electron microscope image of *C. albicans* under the treatment of blank control after 3 h of incubation; (**C**) Effects of NpRS on the membrane potentials of *C. albicans* after 30 min of incubation. All data were expressed as the mean values of triplicate ± standard deviation. ** *p* < 0.01, compared with blank control.

**Figure 4 molecules-28-03098-f004:**
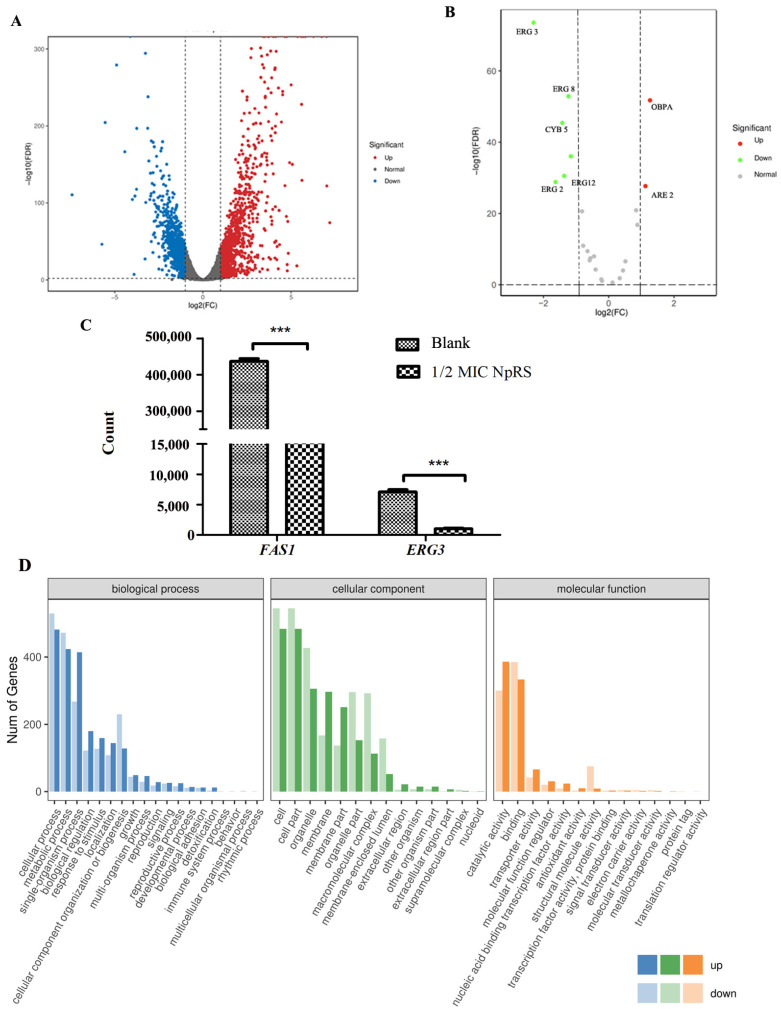
Antifungal mechanism of NpRS explored by transcriptomic analysis. (**A**) Volcano plot of differentially expressed genes of the NpRS treatment group versus the control group. Each dot represents a gene, and the up-regulated and down-regulated genes with the largest fold change are marked in the graph. Fold change is represented on the x-axis, while p adjust is represented on the y-axis. Red and blue indicate differentially up-regulated and differentially down-regulated genes, respectively, while gray dots indicate genes that were not significantly different from the control group; (**B**) Volcano plot of differentially expressed ergosterol-related genes of the NpRS treatment group versus the control group; (**C**) Gene counts of *FAS1* and *ERG3*; (**D**) GO enrichment analysis. Genes were annotated in three main categories: biological process, cellular component, and molecular function. All data are expressed as the mean values of triplicate ± standard deviation. *** *p* < 0.001, compared with blank control.

**Figure 5 molecules-28-03098-f005:**
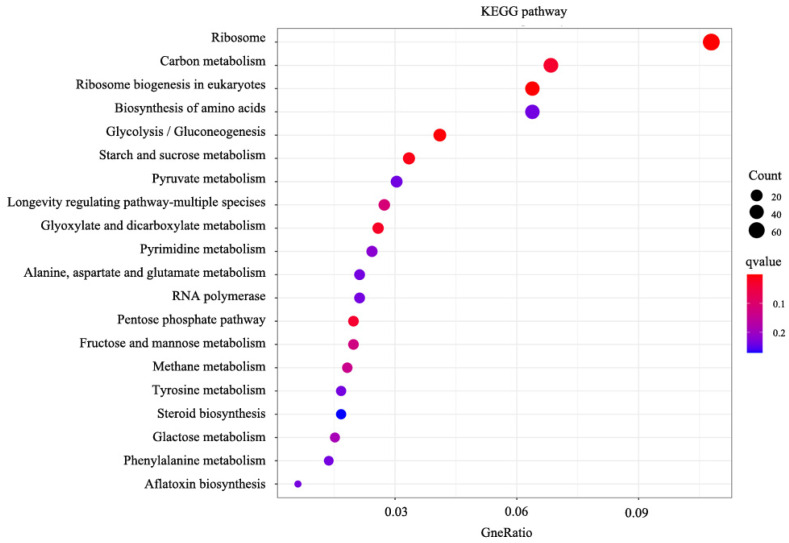
Enrichment scatter diagram of KEGG pathways.

**Figure 6 molecules-28-03098-f006:**
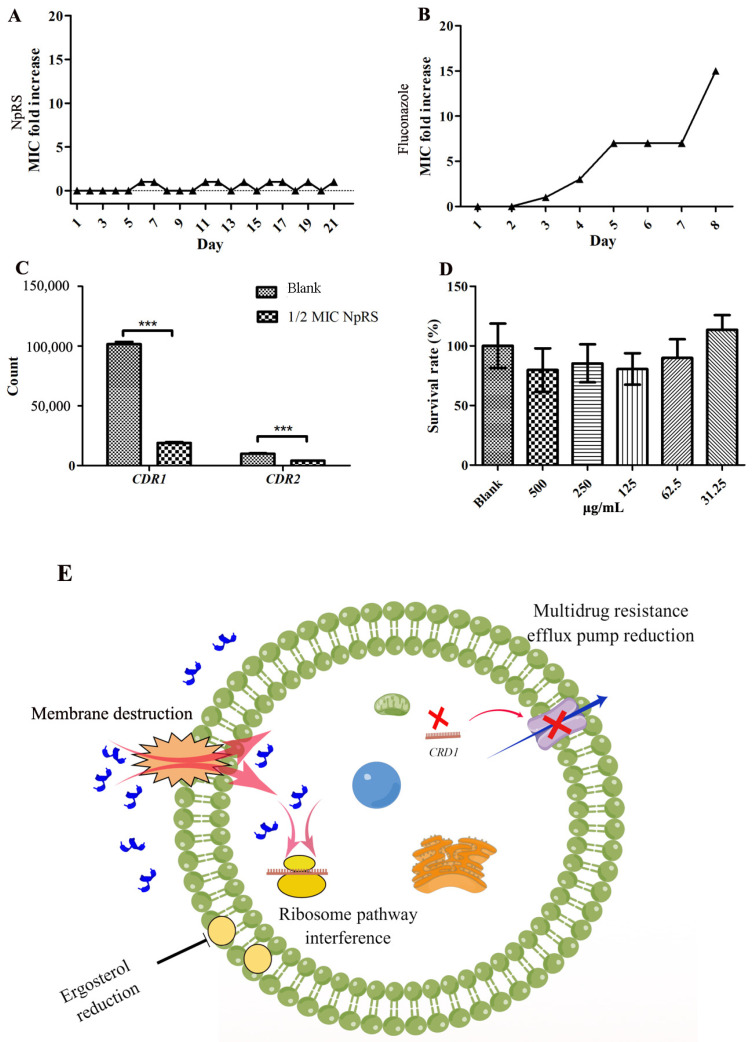
Drug resistance analysis and cell cytotoxicity to mammalian cells. (**A**) Gene counts of *CDR1* and *CDR2*; (**B**) MIC folds increase of fluconazole in 8 days; (**C**) MIC folds increase of NpRS in 21 days; (**D**) Cell cytotoxicity of NpRS to mammalian L6 cells; (**E**) Proposed antifungal mechanisms of NpRS against *C. albicans*. All data were expressed as the mean values of triplicate ± standard deviation. *** *p* < 0.001, compared with blank control.

## Data Availability

Data will be available on request.

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
