# Peer review of "Structural Characterization, Cytotoxicity, and the Antifungal Mechanism of a Novel Peptide Extracted from Garlic (Allium sativa L.)"

_molecules, 2023, doi:10.3390/molecules28073098_

Round 1

Reviewer 1 Report

This article “Structural characterization and the antifungal mechanism of a novel peptide extracted from garlic (Allium Sativa L.)” aimed to discover new antifungal peptides from garlic and identify its structure-activity relationship as well as antimicrobial mechanism. The work is meaningful with sufficient data and in line with readers' interests of Molecules. However, there are still some shortcomings that need to be further improved or explained.

Comments:

Q1. Line 90, >50%, here represents protein content?

Q2. Why did authors choose DEAE-fast flow chromatography?

Q3. Why NMR spectra were not reflected in the main text?

Q4. It is suggested to add more purity and structure infomation of NpRS.

Finally, my main concern is the Antifungal mechanism of NpRS. Whether it is possible that undetected substances exert this activity.

Author Response

Response to the reviewers’ comments

Journal name: Molecules

Manuscript ID: molecules-2293745

Title: Structural characterization, cytotoxicity and the antifungal mechanism of a novel peptide extracted from garlic (Allium Sativa L.)

Dear editor and reviewers:

Thank you very much for providing constructive and profound suggestions for my manuscript entitled “Structural characterization, cytotoxicity and the antifungal mechanism of a novel peptide extracted from garlic (Allium Sativa L.)”. Those comments are very helpful for revising and improving our paper. We have studied the comments carefully and made corrections which we hope meet with approval. The main corrections are marked in the manuscript and the responses to the reviewer’s comments are as follows.

Response to Reviewer 1.

This article “Structural characterization and the antifungal mechanism of a novel peptide extracted from garlic (Allium Sativa L.)” aimed to discover new antifungal peptides from garlic and identify its structure-activity relationship as well as antimicrobial mechanism. The work is meaningful with sufficient data and in line with readers' interests of Molecules. However, there are still some shortcomings that need to be further improved or explained.

Response: Thanks a lot for the comments or advice. The list of responses is as follows.

 Comments:

Question 1. Line 90, >50%, here represents protein content?

Response: Thank you for your question. The >50% here represents the portion of hydrophobic residues in the peptide. The purity of the peptide was >95%, and the purity information was presented in Line 104 and Line 359.

Question 2. Why did authors choose DEAE-fast flow chromatography?

Response: Thank you for your question. DEAE-fast flow chromatography is an anionic ion exchange chromatography. Due to the ideal characteristics of antimicrobial peptides, the peptide with a cationic character was the ideal target of separation. In order to separate the cationic peptide from the extract, DEAE-fast flow chromatography was used to bind the anionic peptides, and the unbounded fraction was acquired for further analysis.

Question 3. Why NMR spectra were not reflected in the main text?

Response: Thank you for your question. NMR analysis included in our study includes 1 H NMR spectrum, 13C NMR spectrum, HSQC NMR spectrum, HMBC NMR spectrum, COSY NMR spectrum, and NOESY NMR spectrum. All the spectra analyses combined resulted in the structure correlations in the manuscript. We think the six spectra presented in the main manuscript would take a lot of pages and the structure correlation was more important for spectrum presentation. Therefore, we presented the spectra in the supplementary files.

Question 4. It is suggested to add more purity and structure infomation of NpRS.

Response: Thank you for your question. The purity of the peptide was >95%, and the purity information was presented in Line 104 and Line 359.

Question 5. Finally, my main concern is the Antifungal mechanism of NpRS. Whether it is possible that undetected substances exert this activity.

Response: Thank you for your question. in our present study, the peptides were identified first and then synthesized for antifungal activity screening. The peptide was synthesized with a purity of over 95%. Therefore, we thought the purity level can exclude the influence of other undetected substances.

We tried our best to improve the manuscript and made some changes in the manuscript. We appreciate for editors/reviewers’ warm work earnestly, and hope that the correction will meet with approval.

Once again, thank you very much for your suggestions. Looking forward to hearing from you.

Thank you and best regards.

Yours sincerely,

Haixia Chen

Reviewer 2 Report

This manuscript is aiming at reporting isolate antifungal peptides from garlic and  evaluate the antimicrobial activities towards fungal pathogen Candida albicans. Besides, the anti-drug resistance property of the peptide was also evaluated. To achieving these  goals, scanning electron microscopy, membrane depolarization assay and transcriptomic  analysis were utilized to explore the underneath function mechanism of the antifungal  peptides, and several genes related with drug resistance were also analyzed for their implications in overall activity evaluation.

Personally, I believe that within MDPI other journals, such as Molecules would be more suitable for this topic

-         Authors did Cytotoxicity of NpRS against mammalian cell, it is must be add in the title. Where is ethical approval?

-         I suggest separate results from Discussion.

-         In introduction 14 references not enough.

-         The English language used is very poor and there are many parts/sentences where the meaning is not clear. Therefore, significant editing needs to be carried out throughout the text to improve it.

-         There are several parts that it seems that the manuscript was never proof read before submission, which contributes to the poor quality of the text

-         Origin on Candida isolates it is must be added in the materials and methods.

-         There are a lot of taxonomic names either of the pathogens of marine sources that need to be in italics. Please edit this throughout.

-         Some compound names have the first letter in capital and some not. Be consistent and don’t capitalise the first letter unless it’s the start of the sentence

-         English throughout the manuscript is very poor even I found spelling mistakes in the abstract, the author should carefully go through the manuscript and correct English grammar and spelling mistakes.

-         References are much more important, the author should modify references and add the specific area references for example the author should cover 10 year work

Author Response

Response to the reviewers’ comments

Journal name: Molecules

Manuscript ID: molecules-2293745

Title: Structural characterization, cytotoxicity and the antifungal mechanism of a novel peptide extracted from garlic (Allium Sativa L.)

Dear editor and reviewers:

Thank you very much for providing constructive and profound suggestions for my manuscript entitled “Structural characterization, cytotoxicity and the antifungal mechanism of a novel peptide extracted from garlic (Allium Sativa L.)”. Those comments are very helpful for revising and improving our paper. We have studied the comments carefully and made corrections which we hope meet with approval. The main corrections are marked in the manuscript and the responses to the reviewer’s comments are as follows.

Response to Reviewer 2.

This manuscript is aiming at reporting isolate antifungal peptides from garlic and evaluate the antimicrobial activities towards fungal pathogen Candida albicans. Besides, the anti-drug resistance property of the peptide was also evaluated. To achieving these goals, scanning electron microscopy, membrane depolarization assay and transcriptomic analysis were utilized to explore the underneath function mechanism of the antifungal peptides, and several genes related with drug resistance were also analyzed for their implications in overall activity evaluation.

Personally, I believe that within MDPI other journals, such as Molecules would be more suitable for this topic

Response: Thanks a lot for the comments or advice. The list of responses is as follows.

Question 1: -Authors did Cytotoxicity of NpRS against mammalian cell, it is must be add in the title. Where is ethical approval?

Response: Thank you for your suggestion. The Cytotoxicity of NpRS against mammalian cells was added in both results and methods sections. In the title cytotoxicity is added. In this study, we did cytotoxicity of NpRS against mammalian cell L6 cell line, which is a kind of rat myoblast cells, and the experiment only studied in cell line but not living animals. According to the Instruction for the Authors, “Methods sections for submissions reporting on research with cell lines should state the origin of any cell lines. An example of Ethical Statements: The HCT116 cell line was obtained from XXXX.” The ethical statement is involved in living animals or experimental sources related to humans. Therefore, the origin of the cell line source was listed in the main text instead of the ethical statement.

Question 2: - I suggest separate results from Discussion.

Response: Thank you for your suggestion. Before the manuscript writing, we carefully read the Instruction for the Authors of Molecules, and the combined results and discussion is acceptable according to the instruction. We carefully considered your suggestion, but the separation requires a new organization of the article structure and language, and the given revision deadline was not enough for this. So, we sincerely ask for your understanding.

Question 3: -In introduction 14 references not enough.

Response: Thank you for your suggestion. The introduction has been enriched with more references within ten years of research.

Question 4:  The English language used is very poor and there are many parts/sentences where the meaning is not clear. Therefore, significant editing needs to be carried out throughout the text to improve it. There are several parts that it seems that the manuscript was never proof read before submission, which contributes to the poor quality of the text.

Response: Thank you for your suggestion. The whole manuscript has been grammar and spelling checked again, and the English expression has been edited by a native speaker for fluency. We sincerely hope that the edited manuscript would match the standard.

Question 5: Origin on Candida isolates it is must be added in the materials and methods.

Response: Thank you for your suggestion. The origin of fungi strain was obtained from the China Center of Industrial Culture Collection (CICC, China). The information was added to the manuscript.

Question 6: There are a lot of taxonomic names either of the pathogens of marine sources that need to be in italics. Please edit this throughout.

Response: Thank you for your suggestion. The taxonomic names have been edited throughout the manuscript. We hope the present manuscript would meet the standard.

Question 7: Some compound names have the first letter in capital and some not. Be consistent and don’t capitalise the first letter unless it’s the start of the sentence

Response: Thank you for your suggestion. The compound name in the manuscript has been reviewed and edited for improvement. We hope the present manuscript would meet the standard.

Question 8: English throughout the manuscript is very poor even I found spelling mistakes in the abstract, the author should carefully go through the manuscript and correct English grammar and spelling mistakes.

Response: Thank you for your suggestion. The whole manuscript has been grammar and spelling checked, and the manuscript has been edited by a native speaker. We sincerely hope the present manuscript would meet the standard.

Question 9: References are much more important, the author should modify references and add the specific area references for example the author should cover 10 year work

Response: Thank you for your suggestion. The references have been enriched and have been added to the manuscript. We systematically reviewed 10 years of work about garlic antifungal activity and also antifungal agents from other sources. We hope the present references would be enough for the manuscript.

We tried our best to improve the manuscript and made some changes in the manuscript. We appreciate for editors/reviewers’ warm work earnestly, and hope that the correction will meet with approval.

Once again, thank you very much for your suggestions. Looking forward to hearing from you.

Thank you and best regards.

Yours sincerely,

Haixia Chen

Reviewer 3 Report

The authors discovered a novel antifungal peptide NpRS from garlic and identified its structure-activity relationship as well as antimicrobial mechanism. The results from this work were valuable for future drug development and garlic market value improvement. Overall, the paper is well-written and the flow of content is acceptable. The manuscript can be accepted after the authors can address following issues after minor revisions:

The format of unit should meet the requirements of the journal. For example, the authors should not use both “ml” and “mL” in the manuscript.

Line 77 “To achieving these goals” should be revised to “In order to achieve these goals”

Line 121 “C. albicans” should be italic.

Line 126, the authors mentioned that “NpRS exhibited significant fungicide activity (P<0.05)”, however there was no significance analysis of the data presented in Figure 1C.

Line 441 “5% CO2 at 37oC” should be revised.

In the section 4. conclusion, the authors should also raise questions and identify areas which need further research.

Author Response

Response to the reviewers’ comments

Journal name: Molecules

Manuscript ID: molecules-2293745

Title: Structural characterization, cytotoxicity and the antifungal mechanism of a novel peptide extracted from garlic (Allium Sativa L.)

Dear editor and reviewers:

Thank you very much for providing constructive and profound suggestions for my manuscript entitled “Structural characterization, cytotoxicity and the antifungal mechanism of a novel peptide extracted from garlic (Allium Sativa L.)”. Those comments are very helpful for revising and improving our paper. We have studied the comments carefully and made corrections which we hope meet with approval. The main corrections are marked in the manuscript and the responses to the reviewer’s comments are as follows.

Response to Reviewer 3.

The authors discovered a novel antifungal peptide NpRS from garlic and identified its structure-activity relationship as well as antimicrobial mechanism. The results from this work were valuable for future drug development and garlic market value improvement. Overall, the paper is well-written and the flow of content is acceptable. The manuscript can be accepted after the authors can address following issues after minor revisions:

Response: Thanks a lot for the comments or advice. The list of responses is as follows.

Question 1: The format of unit should meet the requirements of the journal. For example, the authors should not use both “ml” and “mL” in the manuscript.

Response: Thank you for your suggestion. The writing coherence of the unit has been checked and edited.

Question 2: Line 77 “To achieving these goals” should be revised to “In order to achieve these goals”

Response: Thank you for your suggestion. The English expressions of the manuscript, including “To achieving these goals”, have been checked and edited for fluency.

Question 3: Line 126, the authors mentioned that “NpRS exhibited significant fungicide activity (P<0.05)”, however there was no significance analysis of the data presented in Figure 1C.

Response: Thank you for your question. Actually, there is significant analysis in Figure 1C, but the error bar was shaded by the big curve symbols. The figure has been edited for a better vision of significant analysis.

Question 4: Line 441 “5% CO2 at 37oC” should be revised.

Response: Thank you for your suggestion. The typing error has been edited and revised.

Question 5: In the section 4. conclusion, the authors should also raise questions and identify areas which need further research.

Response: Thank you for your suggestion. The conclusion has been enriched with more perspectives related to antifungal peptide future development and questions about the areas. We hope the present manuscript would meet your standard for publication.

We tried our best to improve the manuscript and made some changes in the manuscript. We appreciate for editors/reviewers’ warm work earnestly, and hope that the correction will meet with approval.

Once again, thank you very much for your suggestions. Looking forward to hearing from you.

Thank you and best regards.

Yours sincerely,

Haixia Chen
